# A qualitative exploration of video-based motor action observation perceptions in patients with chronic low back pain and asymptomatic participants: An interpretative phenomenological analysis

Roy La Touche[1,2,3,4☯], José Vicente León-Hernández[2,3], Prado Silván-Ferrero[5], Encarnación Nouvilas-Pallejá[5], Alba Paris-Alemany[3,4,6☯*], Miguel Ángel Sorrel[7], Joaquín Pardo-Montero[2,3,8]

1 Escuela de Doctorado, Programa de Doctorado en Psicología, Universidad Autónoma de Madrid, Madrid, Spain, 2 Departamento de Fisioterapia, Centro Superior de Estudios Universitarios La Salle, Universidad Autónoma de Madrid, Madrid, Spain, 3 Motion in Brains Research Group, Centro Superior de Estudios Universitarios La Salle, Universidad Autónoma de Madrid, Madrid, Spain, 4 Instituto de Dolor Craneofacial y Neuromusculoesquelético (INDCRAN), Madrid, Spain, 5 Department of Social and Organizational Psychology, National University of Distance Education, Madrid, Spain, 6 Department of Basic Health Sciences, Universidad Rey Juan Carlos, Alcorcón, Spain, 7 Departamento de Psicología Social y Metodología, Universidad Autónoma de Madrid, Madrid, Spain, 8 Hospital La Paz Institute for Health Research (IdiPAZ), Madrid, Spain

☯ These authors contributed equally to this work.
* alba.paris@urjc.es

## Abstract

### Background

Video-based action observation (AO) of exercise/motor-action content is increasingly delivered via social media. This expands reach and ecological validity but may shape motor simulation, perceived safety, and engagement. How people with chronic low back pain (CLBP) interpret and intend to use such videos remains underexplored.

### Methods

We conducted an interpretative phenomenological analysis (IPA) of semi-structured interviews with purposively sampled adults (n = 20; 10 CLBP, 10 asymptomatic). Interviews probed perceptions of exercise/motor-action AO videos drawn from common platforms. Analysis followed IPA procedures with iterative coding, constant comparison, and team reflexivity, and is reported according to COREQ.

### Results

Three interrelated themes were identified: (1) Emotional & motivational impact: CLBP participants frequently appraised bending, load and fast tempo as threatening and described protective avoidance rules. Motivation was present in both groups when videos felt safe

**Data availability statement:** All relevant data are within the paper and its Supporting information file.

**Funding:** JPM received a funding from Centro Superior de Estudios Universitarios La Salle with the grant number GI2012002C. This funder did not play any role in the study design, data collection or analysis, decision to publish, or preparation of the manuscript. APA and RL were funded by the Illustrious Professional College of Physiotherapists of the Community of Madrid through the IV Call for Research Grants in Physiotherapy to support the EMFRA project (A98145 URJC).

**Competing interests:** The authors have declared that no competing interests exist.

and adaptable; (2) Self-assessment of physical capacity: Perceived competence increased when videos provided graded options and explicit safety cues. Anticipated task demand decreased with appropriate pacing/tempo, egocentric viewpoint, and credible modeling; (3) Cognitive movement assessment: Viewers attended to posture, tempo, breathing and error-avoidance cues. Action comprehension faltered when instructions were dense/fast or goals were unclear. Judgments about delivery (goal clarity, safety cues, pacing, viewpoint, modeling fidelity) shaped internal rehearsal and willingness to attempt.

## Conclusions

Individuals with and without CLBP perceive social-media–delivered exercise AO as useful when videos are tailored (graded options, clear safety messaging, appropriate pacing/viewpoint) and contextualized to pain-related concerns and digital literacy. These insights inform clinically oriented AO exercise-video libraries and implementation strategies (e.g., curated playlists, level-tagging, therapist-mediated briefing) to enhance acceptability and adherence in CLBP rehabilitation.

## Introduction

Chronic pain is a major public health problem with substantial individual and societal impact due to its prevalence, persistence, and multidimensional consequences [1]. Within musculoskeletal conditions, chronic low back pain (CLBP) is particularly disabling—affecting roughly a quarter of adults—and is associated with functional limitations, reduced participation, and variable levels of pain-related disability [2–4].

Action observation (AO) training—situated within movement representation techniques—uses the systematic observation of exercise/motor actions from egocentric or allocentric perspectives, directing attention to key task elements and encouraging analytic processing of the movement [5,6].

AO is grounded in social learning theory, whereby attention, retention, motor reproduction, and motivation support vicarious acquisition and refinement of motor skills; self-efficacy is central to whether individuals attempt and persist with the observed behaviors [7–11].

From a neurophysiological standpoint, observing actions engages networks implicated in planning, preparation, and execution and can increase corticospinal excitability and autonomic responses consistent with motor simulation and motor resonance [12–17]. Although the mirror neuron system has been widely discussed [18,19], contemporary accounts emphasize that observational learning relies on multiple interacting systems, that responses are context sensitive, and that familiarity and perceived intention modulate neural activity during action perception [20–23].

Clinically, AO has shown hypoalgesic or pain-reducing effects in asymptomatic participants, in chronic pain populations, and even in acute postoperative settings [24–28]. However, most prior work has focused on what is observed and the downstream effects, rather than how people actually perceive and make sense of the observed movements—whether they find them comprehensible, safe, threatening, or motivational; how instructional features (e.g., pacing, viewpoint, modeling fidelity) shape imagined

execution; and how these appraisals influence willingness to try exercises. This gap is clinically salient. Pain can alter automatic imitation and processing of others' actions [29,30], and related paradigms using photos or videos to probe fear of movement in musculoskeletal pain—and neuroimaging in CLBP—have tended to emphasize fear/difficulty rather than the broader interpretative and evaluative dimensions of AO [31–35].

At the same time, social-media and video-sharing platforms (e.g., YouTube) have become prominent channels for disseminating exercise content. These platforms can expand reach, enhance ecological validity, and potentially improve digital health literacy and engagement with low back pain programs [36,37]. Platform conventions (e.g., titles, thumbnails, on-screen cues), perceived credibility, comment sections, and parasocial dynamics with creators may all shape trust, risk appraisal, and adherence intentions—variables directly relevant to AO and particularly pertinent in CLBP, where pain-related fear and low self-efficacy commonly hinder exercise participation. Emerging work suggests that tailored online routines can support motivation and sustained participation, including in populations with heavy work demands [38]. Yet little is known about how individuals with CLBP and those without pain interpret, evaluate, and intend to use social-media exercise/motor-action AO videos.

Given these gaps, a qualitative design is warranted to capture the lived experience and meaning-making processes that underlie acceptance and use of AO videos in real-world digital contexts. Interpretative phenomenological analysis (IPA) is well suited to this task because it examines how participants make sense of significant experiences, attending both to idiographic detail and to shared patterns across cases. Understanding these interpretative processes can inform the design of AO video resources that are not only biomechanically appropriate but also perceived as comprehensible, credible, and safe—conditions likely to influence adherence and clinical outcomes.

Aim and guiding questions. This study used IPA to explore how adults with CLBP and asymptomatic adults perceive social-media exercise/motor-action AO videos. We asked: (i) How do participants appraise the utility and safety of the observed exercises, and what fears or facilitators emerge? (ii) Which instructional (e.g., clarity of goals, safety cues) and representational (e.g., pacing/tempo, point-of-view, modeling fidelity) features shape motor simulation and imagined execution? and (iii) Which contextual factors (e.g., prior pain experiences, self-efficacy, digital literacy and platform conventions) influence trust, willingness to attempt movements, and perceived relevance for adherence? By centering these experiential dimensions, the study aims to generate practice-oriented insights to guide the development and clinical implementation of tailored AO exercise-video libraries and therapist-mediated strategies for CLBP rehabilitation.

## Methods

### Study design

This study employs an exploratory qualitative approach and was designed in accordance with the Consolidated Criteria for Reporting Qualitative Research [39] and the Standards for Reporting Qualitative Research [40]. These criteria were specified and applied at each stage of the study, ensuring completeness and transparency in data reporting and analysis. The complete semi-structured interview guide is provided in Supplementary S1 File.

A qualitative, cross-sectional methodology was adopted using Interpretative Phenomenological Analysis (IPA), as described by Smith [41]. This flexible and versatile qualitative method assesses the meaning that individuals attribute to their lived experiences, allowing the exploration, description, interpretation, and contextualization of these experiences from the perspective of a small number of participants [42–44]. We focused specifically on perceptions of social-media exercise/motor-action videos used for action observation (AO), and reporting follows COREQ.

### Ethics

The study was approved by the bioethics committee of the La Salle University Center for Advanced Studies (CSEULS-PI-005/2020). The procedures performed in this study complied with the ethical principles of the Declaration of Helsinki for research on human subjects [45].

Prior to the semi-structured interviews, participants provided written informed consent, agreeing to participate, allowing audio recording, and acknowledging confidentiality terms that ensured their anonymity throughout the study. Specific measures for data anonymization included removing any personal identifiers from interview transcripts (e.g., names, contact details, contextual references) and replacing them with a unique alphanumeric code assigned to each participant. These codes were used at all stages of the analysis to maintain confidentiality. Additionally, all audio recordings and transcripts were stored on a password-protected server, accessible only to the core research team. Participants were informed that they could request and review their transcripts for accuracy if they wished.

## Participants

Patients with chronic nonspecific low back pain (CNSLBP) were recruited from two rehabilitation clinics in Madrid. The inclusion criteria were: a) having low back pain for at least 6 months; b) nonspecific low back pain; c) pain intensity of 3 or higher on the numerical pain scale; d) age between 18 and 65 years; e) having undergone a therapeutic exercise treatment that included instruction and learning through the observation of actions via videos; and (f) having accessed these exercise videos through social media at least three times in the last two months. The exclusion criteria included: a) having undergone back surgery in the past year; b) having specific spinal diseases such as malignant or inflammatory diseases of the joints and bones; c) having experienced spinal trauma in the past year; d) having infectious or tumorous diseases; e) presenting uncorrected visual impairments and/or cognitive disabilities that would prevent adequate understanding of audiovisual material; f) having difficulties in comprehension or communication; and g) having insufficient knowledge of Spanish to follow the instructions related to the collected variables. To enhance heterogeneity, purposive sampling sought variation in sex, age strata, duration/severity of CLBP, and level of digital literacy/usage of social-media exercise content.

For the recruitment of asymptomatic participants, a sample was drawn from the CSEULS campus through social media postings, faculty announcements, direct contact with participants, and posters. The participants recruited included students, administrative or teaching staff, and others in the area who wished to participate in the study. Inclusion criteria for asymptomatic participants were (a) no pathology or pain in the last 3 months; (b) age between 18 and 65 years; (c) previous experience in performing exercise with instruction and learning the exercises through videos; and (f) having accessed these exercise videos through social media at least three times in the last two months. The exclusion criteria for this group were (a) having uncorrected visual disabilities and/or cognitive disabilities that prevented adequate comprehension of the audiovisual material; (b) having difficulties in comprehension or communication; or (c) having insufficient knowledge of Spanish to follow the instructions of the variables collected. All participants provided written informed consent prior to the start of the research.

## Study context

To ensure an atmosphere of confidentiality and comfort, interviews and data collection were conducted in private offices within the facilities of the clinics where the study took place. During the interview sessions, only the researcher and the participant were present, without the intervention of third parties. This ensured an atmosphere of trust that facilitated openness and honesty in participants' responses. When necessary, scheduling accommodated participant preference to minimize burden.

## Procedure

Semi-structured in-depth interviews were employed, applying an interview guide developed on the basis of research team meetings and the scientific literature on action observation in patients with pain. The interview protocol was designed according to the criteria set out by DeJonckheere and Vaughn [46].

The interview questions were tested on a pilot group of patients with CNSLBP not included in the study, allowing for adjustments such as simplifying certain terms and including additional questions to further explore related emotional experiences. The finalized guide (items and probes) appears verbatim in Supplementary S1 File. Pilot feedback also led to adding probes on explicit safety cues, graded options, pacing/tempo, and point-of-view (egocentric/allocentric). Pilot data were not included in the analysis. These additions are marked in the guide and reflected in the versioning notes of the audit trail (S2 File).

Additionally, participants were asked the following question to assess their engagement with exercise videos: "Which social media or online platform did you use to watch the exercise videos, and how many times did you do so?". This preliminary phase eliminated the need for repeated interviews.

Audio recording devices were employed to capture the interviews, with the informed consent of the participants. Interviews lasted between 29 and 55 minutes, with an average length of approximately 42 minutes. Recordings were transcribed literally.

To ensure fuller responses or in cases of brief responses from participants, follow-up questions were used, such as "Could you expand on that?", "What do you mean by that?", or "Could you explain more about...?" These additional questions were applied flexibly to gain a deeper understanding of the patients' experiences.

During the interviews, the researcher took field notes to ensure full coverage of the topics discussed. Immediately after each interview, reflective notes were taken on the relevance of the topics of discussion to facilitate further analysis. This notebook was consulted during the transcription and analysis phases to incorporate reflective comments on the researcher's insights and the interpretative process. Although interview transcripts were not automatically returned to participants for comments or corrections, participants were informed that they could request and review their transcripts if they wished. Reflexive notes were revisited during coding meetings to make explicit assumptions and to support consensus decisions.

## Sample size

A non-probability purposive sampling method was employed to select participants for both phases of the study. This approach is based on choosing participants according to the research question and the objectives of the study [47]. The sample consisted of 10 patients with chronic low back pain and 10 asymptomatic participants. This sample size is in line with recommendations that suggest including 2–12 participants for interpretative phenomenological analysis [48]. The final size (n = 20) was further justified by information power and the idiographic depth sought in IPA. Saturation was operationalized as the absence of novel subthemes toward the end of data collection, confirmed by team consensus.

## Qualitative analysis

The qualitative analysis of the data was carried out in seven distinct phases to ensure accurate and consistent interpretation. A step-by-step description of the seven-phase analytic flow, with "evidence hooks" and versioning, is summarized in the Audit Trail (Supplementary S2 File).

In the first phase, transcripts were read and re-read to achieve deep immersion in the data. This was complemented by listening repeatedly to the interview recordings, capturing the emotions, tone, and emphasis of the participants

In the second stage, exploratory notes and codes were recorded both manually (during the reading) and digitally through ATLAS.ti. [49,50]. This software was selected for its capacity to manage large volumes of qualitative data and to facilitate multi-stage coding. Operational boundary rules (e.g., Evaluation of instruction vs Action comprehension), concrete discrepancy examples, and the dated changelog of definitions are documented in Supplementary S2 File.

The third phase involved generating experiential statements by identifying the most salient elements from the previous steps and constructing conceptual meanings based on both transcript content and researchers' interpretations. In the fourth phase, the emerging connections and patterns among these experiential statements were examined in light of the study's objectives. The fifth phase mapped and named these connections, resulting in the final coding scheme. The sixth

phase repeated these processes for each interview, and in the seventh phase, individual experiential themes were refined to produce a synthesis of group-level themes. The full codebook (themes, sub-themes, analytic definitions, exemplar quotations, and indicative counts by group) is provided in Supplementary information S1 Table.

To ensure reliability and validity within this analytic framework, data triangulation and investigator triangulation were employed. Specifically, two members of the research team coded the transcripts independently, and any inconsistencies in codes or interpretations were resolved through discussion to achieve consensus, with moderation by a third investigator when needed. This multi-researcher approach helped minimize potential biases and enhance credibility. Additionally, reflexivity was maintained through field notes and ongoing critical reflection, allowing the team to acknowledge and manage their own assumptions and perspectives. Coding proceeded primarily inductively within the IPA framework; limited deductive sensitization (e.g., fear of movement, motor simulation, instructional features) was used only to orient early memoing, with final themes emerging from iterative, data-driven interpretation. Independent dual coding was followed by scheduled consensus meetings (third-researcher moderation if required). All qualitative data were managed and coded in ATLAS.ti (Scientific Software Development GmbH).

Data saturation was considered achieved when no new themes or significant ideas emerged. This point was established through consensus among the researchers regarding the sufficiency and redundancy of the information collected. Themes were identified inductively from the data, facilitating an informed and authentic interpretation of participants' experiences. Lastly, the selection of illustrative quotes for reporting the findings was made by consensus within the research team, ensuring that excerpts represented both typical and revealing perspectives. ATLAS.ti software was used for data management and analysis throughout the process [49].

### Positionality and reflexivity

The interdisciplinary team comprised clinicians and researchers with prior experience in CLBP rehabilitation and AO. Anticipated assumptions (e.g., expectations regarding exercise safety and AO utility) were addressed through reflexive journaling, peer-debriefing sessions, iterative memoing, and negotiated consensus during analysis. Decisions about code boundaries and theme formation were documented and revisited across coding meetings. Illustrative memos (e.g., on safety messaging and viewpoint/pacing) are excerpted in Supplementary S2 File.

### Trustworthiness

To enhance trustworthiness, we applied a focused set of procedures: (i) Credibility—independent dual coding with consensus meetings; (ii) Dependability—maintenance of an analytic decision log/audit trail documenting codebook evolution and theme clustering; (iii) Confirmability—reflexive memos and retention of the evolving codebook, code-application examples, and theme maps; and (iv) Transferability—maximum-variation purposive sampling and detailed description of setting, participants, and digital context to support informed judgments about applicability. Details of credibility checks, consensus procedures, reflexive memos, and storage of intermediate maps are compiled in the Audit Trail (Supplementary S2 File).

## Results

### Sociodemographic data from the qualitative analysis

The qualitative analysis was conducted with 20 participants (10 patients with CLBP and 10 asymptomatic participants), adult women and men of similar age. Of the total participants included in the study, 16 (80%) had performed action observation of exercise videos on YouTube, whereas the remaining 4 (20%) utilized Instagram for this purpose. Overall, the frequency with which participants reported viewing these exercise videos ranged from 5 to 10 occasions. To streamline presentation, we report only core variables in Table 1. Table 1 shows the sociodemographic characteristics of the participants who took part in the qualitative study. Table 2 provides a compact summary (theme, sub-theme, analytic definition,

**Table 1. Characteristics of the qualitative study sample.**

**Patients with Chronic Low Back Pain**

| Identification: Pain Patients (PP) -Asymptomatic patients (AP) | Sex (M/F; Male/Female) | Age | Pain Duration (years) | BMI | Marital Status | Employment Status | Educational Level |
|---|---|---|---|---|---|---|---|
| PP1 | Woman | 44 | 7 | 25.8 | Married | Sick leave | Secondary |
| PP2 | Woman | 46 | 12 | 28.6 | Divorced | Active worker | University Studies |
| PP3 | Woman | 32 | 6 | 24.7 | Single | Unemployed | University Studies |
| PP4 | Woman | 26 | 4 | 26.5 | Single | Active worker | Vocational training |
| PP5 | Woman | 45 | 15 | 27.7 | Married | Sick leave | University Studies |
| PP6 | Man | 56 | 10 | 23.5 | Divorced | Unemployed | Secondary |
| PP6 | Woman | 51 | 7 | 27.7 | Married | Active worker | Vocational training |
| PP7 | Man | 64 | 14 | 21.4 | Married | Active worker | University Studies |
| PP8 | Man | 25 | 3 | 20.4 | Single | Sick leave | Vocational training |
| PP9 | Woman | 39 | 19 | 34.1 | Married | Sick leave | Secondary |
| PP10 | Woman | 62 | 16 | 22.3 | Married | Active worker | University Studies |
| **Asymptomatic Participants** | | | | | | | |
| AP1 | Man | 57 | – | 19.4 | Married | Active worker | University Studies |
| AP2 | Woman | 43 | – | 22.3 | Married | Active worker | University Studies |
| AP3 | Woman | 32 | – | 20.6 | Married | Active worker | University Studies |
| AP4 | Man | 22 | – | 25.8 | Single | Student | University Studies |
| AP5 | Woman | 60 | – | 21.3 | Married | Sick leave | Secondary |
| AP6 | Man | 34 | – | 18.5 | Single | Unemployed | Vocational training |
| AP7 | Man | 52 | – | 21.9 | Single | Active worker | University Studies |
| AP8 | Woman | 41 | – | 19.8 | Divorced | Active worker | Vocational training |
| AP9 | Woman | 23 | – | 20.9 | Single | Student | University Studies |
| AP10 | Woman | 38 | – | 24.9 | Married | Active worker | Secondary |

BMI: Body mass index.

Note: ID labels were audited to ensure unique identifiers (PP1–PP10; AP1–AP10).

exemplar quotation, and indicative counts). A compact codebook with sub-themes, analytic definitions, exemplar quotations, and indicative counts (CLBP/AP), plus the thematic map, is available in Supplementary File S1 Fig.

We report simple indicative counts where helpful (e.g., "n/10 CLBP; n/10 asymptomatic") to convey prevalence while maintaining a qualitative emphasis. Subthemes are presented in order of observed prevalence/centrality across cases.

## Results of the interpretative phenomenological analysis

Three main themes and several sub-themes were derived from the IPA (see Table 2): Emotional and Motivational Impact, Self-Assessment of Physical Capacity, and Cognitive Assessment of Movement. Each theme is introduced with a brief link to the guiding research questions and illustrated with a small number of non-redundant quotations selected to exemplify the analytic claims. Detailed descriptions and in-depth analyses of each theme and sub-theme are presented below, including literal quotes from participants to exemplify the analysis, as well as comparisons between the patients with pain (PPs) and the asymptomatic participants (APs). Additional exemplar quotes and cross-case notes are provided in Supplementary S1 Table.

## Emotional and motivational impact

This topic focused on describing how watching exercise videos influenced participants' emotions and motivation, highlighting the importance of this visual medium in the perception of and emotional response to physical activity. The visualization

**Table 2. Summary of themes, subthemes, analytic definitions, exemplar quotations, and indicative counts.**

| Theme | Sub-theme | Short analytic definition | Exemplar quotation (ID) | CLBP (n/10) | Asymp-tomatic (n/10) |
|---|---|---|---|---|---|
| Emotional & Motivational Impact | Fear of movement | Anticipatory pain/threat appraisals (flexion, load, tempo) shaping willingness to try exercises. | PP10. "Watching some exercises, I worry I'll do them wrong and flare my pain." | 6/10 | 1/10 |
| | Avoidance beliefs/behaviors | Protective rules and avoidance scripts triggered by perceived risk. | PP7. "If a movement looks risky, I'd rather skip it." | 8/10 | 3/10 |
| | Motivation for movement | Encouragement/hope contingent on perceived safety and adaptability. | AP10. "Seeing clear demonstrations makes me want to try them." | 9/10 | 10/10 |
| Self-Assessment of Physical Capacity | Competence | Confidence to execute safely/effectively, often when graded options/safety cues are present. | PP1. "I can adapt the exercise, so it doesn't worsen my pain—that gives me confidence." | 4/10 | 5/10 |
| | Perceived demand | Anticipated difficulty/workload based on pace, viewpoint (egocentric), and modeling fidelity. | PP4. "Some moves seem beyond me; I'd need an easier version." | 7/10 | 6/10 |
| Cognitive Movement Assessment | Attention to activity | Focus on posture, tempo, breathing, and error-avoidance cues during viewing. | PP9. "I watch posture and tempo closely to understand the movement." | 6/10 | 7/10 |
| | Action comprehension | Understanding of mechanics/sequence; ability to run an internal simulation. | PP6. "Without step-by-step explanation, I don't fully understand how to do it." | 5/10 | 8/10 |
| | Evaluation of instruction | Judgments about delivery (clarity of goals, explicit safety cues, pacing/tempo, viewpoint, modeling). | PP3. "Beginner versions and slower pace would help me follow along." | 4/10 | 6/10 |
| | Familiarity with the motor action | Prior exposure/experience facilitating recognition and simulation. | AP9. "Because I've done something similar, it's easier to picture and try." | 4/10 | 5/10 |

of movement through video can evoke emotions that could in turn influence willingness to participate in exercise, influencing the effectiveness of exercise programs:

PP3: "I am worried about watching the videos and then doing the movements incorrectly and causing more pain."

PP7: "I pay particular attention to movements that require me to bend my back, I have to be careful with these exercises, I don't find them very suitable for this pain."

Fear of bending/load was commonly referenced by CLBP participants (6/10) and less frequently by asymptomatic participants (1/10).

**Sub-theme: Fear of movement.** Fear of movement is especially relevant when participants observe exercises that they perceive as potentially painful or harmful. This fear is deeply rooted in anticipation of pain based on previous experiences or observation of movements that appear challenging or dangerous. This fear might be related to pain anticipation behaviors, which could arise from previous experiences or from observing movements that the individual might assess as very challenging or dangerous:

PP5: "I see some exercises and I am afraid to do them because I know it will hurt."

PP10: "I feel worried when I watch the videos and then do the movements incorrectly and that it might cause me more pain."

In our cross-case reading, fear clustered around perceived spinal flexion, axial loading, and rapid tempo (6/10 CLBP; 1/10 asymptomatic).

The visualization of movement through video can evoke emotions that may, in turn, shape willingness to participate in exercise and the perceived effectiveness of exercise programs.

In addition, repeated exposure to visualization of the exercises or movements could be a therapeutic measure to reduce the fear of movement:

PP6: "It would be good to look for videos that offer a lot of modifications and clear explanations, especially designed for people with chronic pain. The good thing is that as you watch more exercises you become less afraid to do them."

**Sub-theme: Avoidance behaviors and beliefs.** Avoidance of certain movements can be directly influenced by how the exercises are presented in the videos. If the movements appear too complex or risky, it could encourage avoidance beliefs that limit participation:

PP1: "The most correct way to do the exercise is important to avoid hurting myself."

PP6: "I pay close attention to movements that might cause me pain so that I don't do them."

PP7: "I pay particular attention to movements that require me to bend my back, I have to be careful with these exercises, I don't find them very suitable for this pain."

Excessive caution can result in a reduction in the physical activity needed to maintain or improve musculoskeletal function, encouraging cycles of inactivity. This subtheme was more frequent among CLBP participants (8/10) than among asymptomatic participants (3/10).

**Sub-theme: Movement motivation.** Motivation to participate in exercise can be significantly affected by how individuals perceive their own ability and the potential benefit of the movements observed in the videos:

PP5: "The videos help, but I'm not sure that they help in all occasions."

PP8: "I was happy to think that these exercises could be a solution to my problem."

For the APs, the videos could have acted as a motivational tool by demonstrating the effectiveness of the exercises. For the PPs, motivation might have depended more on the perceived safety and adaptability of the exercises to their specific needs:

AP10: "I felt well motivated because the exercises in the videos were easy for me."

PP2: "At first I felt insecure, but with practice, I gained more confidence."

PP7: "It didn't go too well at first, because some movements I tried to do caused me pain, but I must also admit that I found some other exercise videos that I could do with very little discomfort."

Positive motivational statements were common in both groups, often contingent on perceived safety and adaptability (9/10 CLBP; 10/10 asymptomatic).

## Self-assessment of physical capacity

This theme was based on how participants perceived and evaluated their own ability to perform the exercises presented in the videos. Self-assessment of physical ability can influence confidence, motivation, and willingness to participate in physical activities. It is possible that this perception of physical ability is established by factors such as previous experiences, current health status, and understanding of the instructions.

**Sub-theme: Competence.** Perceived competence refers to confidence in the ability to execute exercises effectively and safely. In the context of this research, we explored how this perception influenced participants' willingness to engage in exercises visualized through videos, and how it differed between PPs and APs:

AP4: "My first impression was that the videos were well done and that the exercises seemed easy and useful."

PP1: "I thought the exercises were adequate and I could find a way to do the exercises without it hurting me more."

This text extract shows a positive perception of competence conditioned by the adaptability of the exercises. Confidence arose when they felt that they could perform the activities without exacerbating their pain.

The PPs showed confidence when they felt that they could adapt the exercises to their needs, which increased their perception of competence. The APs generally perceived the exercises as accessible, reinforcing their confidence. However, the perception of competence could fluctuate in both groups depending on the complexity of the exercises and the clarity of the instructions.

Reports of adjustable competence increased when videos presented clear safety cues and graded options (4/10 CLBP; 5/10 asymptomatic).

**Subtheme: Expected perceived demand.** Expected perceived demand refers to how participants assessed the difficulty and physical demands of the visualized exercises, as well as their anticipated ability to perform them. This perception can influence the confidence to perform the activity and the willingness to initiate a particular movement or exercise.

Participants expressed a variety of perceptions about the demands of the exercises, possibly influenced by their physical conditions, previous experiences, and personal expectations:

AP8: "I was curious to see if I could do those exercises."

PP4: "They looked simple, but I was worried if I could really do those movements without more pain."

AP2: "I thought from watching the exercises that they were very, very easy for my physical condition at the time."

AP9: "I thought the videos were well explained and would be easy to follow."

The APs perceived the videos as a tool that gave them confidence, possibly because they allowed them to anticipate the difficulty of the exercises and mentally prepare for them. Pre-viewing might reduce uncertainty and increase confidence in performance:

AP8: "When I watch exercise videos it gives me confidence."

AP1: "Some exercises seemed too advanced for me. However, seeing how they were performed step by step in the videos I gained a little more confidence."

Anticipated difficulty frequently decreased when pacing, viewpoint (egocentric), and modeling fidelity were perceived as adequate (7/10 CLBP; 6/10 asymptomatic).

In the case of the PPs, pain could have influenced the perception of demand. Participants might have been generating an association between the difficulty of the movements and the exacerbation of the pain; on the other hand, although the exercises might have been evaluated as simple, there could be an underlying concern about the impact they might have on their painful condition. The expected perceived demand is not only based on technical difficulty, but also on the possible physical consequences:

PP10: "When I watched some of the exercises on video, I found it difficult to do exercises that require lying on your back."

PP5: "I felt a bit insecure because I thought the movements would be too difficult for my pain; it always happens to me that when exercises are too difficult, they are worse for me because it hurts more afterwards."

PP4: "The exercises that required bending down and lifting things, I found them difficult and not for me."

**Cognitive movement assessment**

This topic addresses how participants processed and understood information related to the visualized exercises. It includes aspects such as attention to detail, understanding of instructions, and familiarity with the movements, all of which could have influenced the interpretation of the visualized movement and its possible execution.

**Sub-theme: Attention to the activity.** Attention to the activity refers to the degree of concentration that participants devoted to technical details and specific aspects of the exercises during their observation. This attentional focus can be an essential complement in the process of observational learning and replication of the visualized exercises:

AP3: "I pay attention to the correct posture during each movement to make sure that I will do them well."

PP9: "I carefully observe all movements to understand how the exercise is performed."

AP1: "I pay attention to the proper breathing that should accompany each exercise."

AP6: "It is essential to watch the speed at which the exercises are performed to avoid doing them too fast."

AP9: "The details of the posture, or for example the speed, are the things that interest me the most."

The data revealed that both PPs and APs paid considerable attention to the technical details of the exercises when observing them. However, the motivations for and specific aspects to which they devoted these attentional resources differed between the groups. Attention frequently focused on posture, tempo, breathing, and error-avoidance cues (6/10 CLBP; 7/10 asymptomatic).

**Sub-theme: Action comprehension.** Action comprehension refers to the cognitive processing by which participants interpreted and assimilated the purpose, mechanics, and sequence of the exercises presented, especially through visual media such as videos. This sub-theme identifies how understanding of an action can directly influences the ability to learn the movement and the intention to perform it:

AP7: "Watching the videos of the exercises gives me clues on how to perform them."

PP6: "When I watch the videos, sometimes I feel a bit lost, I don't always understand how to do the movements just by watching."

The PPs could experience difficulties in fully understanding the exercises, especially if adaptations were not provided. They valued detailed explanations and modifications that allowed them to adjust the exercises to their abilities. APs used the videos to better understand the technique and improve their execution:

PP9: "I try to think how to perform the movement, but adapting it to my illness."

PP2: "It is very important to see the different variations an exercise can have for people with pain as me".

AP9: "My attention goes to understanding how the exercise should be performed"

**Sub-theme: Evaluation of instruction.** Evaluation of instruction refers to the process by which participants analyzed the delivery quality—goal clarity, explicit safety cues, pacing/tempo, viewpoint, and modeling fidelity. Operational distinction from "Action Comprehension": the former concerns the delivery (clarity of goals, explicit safety cues, pacing/tempo, modeling fidelity, viewpoint), whereas the latter concerns the viewer's understanding and internal simulation of movement mechanics. In boundary cases where unclear delivery leads to misunderstanding, we coded the primary locus (instruction vs. comprehension) and noted the linkage during cross-case synthesis. Judgments about instructional delivery (clarity of goals, explicit safety cues, pacing/tempo, viewpoint, modeling) were reported across both groups (4/10 CLBP; 6/10 asymptomatic).

PP6: "I like it when there is advice on how to perform the exercises in case of pain."

AP10: "I like to see if there are indications on the correct execution for each exercise."

The quality and clarity of instructions are important for both groups. The PPs needed adaptations and advice specific to their condition, whereas the APs valued flexibility and the possibility to adjust the pace of learning. Lack of additional directions or detailed explanations could be a barrier for both:

PP3: "I wish the videos were slower or had versions for beginner users."

AP5: "I prefer the videos because I can control the pace and review the movements, plus I don't like gyms too much and I can do the exercises at home with the help of the videos."

**Sub-theme: Familiarity with the motor action.** Participants who recognized previously practiced or occupationally familiar movements described easier mental rehearsal and higher willingness to attempt the exercise; unfamiliar sequences, in contrast, prompted caution and rewatching. Familiarity often interacted with instructional clarity to facilitate action comprehension (4/10 CLBP; 5/10 asymptomatic).

AP9: "Because I'd done something similar before, it was easier to picture how to do it."

PP5: "If I haven't seen it before, I need to replay it a couple of times to be sure."

Overall, the three themes converge on the idea that instructional quality (delivery features) and representational choices (pace, viewpoint, modeling fidelity) shape action comprehension and perceived safety, while contextual moderators (pain history, self-efficacy, digital literacy and platform conventions) calibrate trust and willingness to trial movements, particularly among individuals with CLBP.

## Discussion

This IPA examined how adults with and without CLBP make sense of exercise videos used for AO. Rather than treating videos as neutral stimuli, participants construed them as socially and clinically meaningful artifacts that signal safety, competence, and credibility. Three interpretative threads cut across accounts: (1) threat appraisal versus opportunity

appraisal, (2) the role of instructional delivery in enabling understanding, and (3) contextual moderators of uptake (prior pain experiences, self-efficacy, and digital literacy). Below we discuss each contribution and its clinical implications.

Briefly, the analysis converged on three themes—emotional–motivational responses, self-appraisal of physical capacity, and cognitive appraisal of movement—with interlinked subthemes. Participants with CLBP more often read videos through a threat-management lens (fear of bending/load, protective rules, cautious trialing), whereas asymptomatic participants tended to interpret them as opportunities for skill and activity. Motivation emerged in both groups when delivery signaled safety and personal relevance. Perceived competence strengthened when videos offered graded options, explicit safety cues, and credible modeling, yet anticipated task demand could temper willingness to try. Across accounts, viewers attended to posture, tempo, and error-avoidance cues; action comprehension faltered when instructions were dense, fast, or ambiguously framed. Familiarity with the motor action and clear instructional structure helped bridge observation to intended enactment. These qualitative patterns orient the interpretations and practice implications that follow.

## Threat appraisal versus opportunity appraisal

Across accounts, CLBP participants framed AO through a vigilance–threat lens, especially when videos displayed movement features they associated with flare-ups (e.g., bending, loading, fast tempo). This pattern aligns with contemporary extensions of fear-avoidance perspectives in musculoskeletal pain, where anticipatory threat and protective rules constrain activity even before movement occurs [51,52]. This emotional response appears to be mediated by anticipation of pain and previous negative experiences with movement [53,54]. It also resonates with evidence that pain can perturb action perception/simulation, altering automatic imitation and motor resonance [30].

Our data add that video properties can serve as threat or safety cues within viewers' interpretative schemas. Conversely, many participants—CLBP and asymptomatic—experienced AO as an opportunity for vicarious learning and preparation, consistent with social-cognitive models in which observational experience can enhance perceived capability and motivation [11,55].

In digitally delivered care, recent large trials indicate that supported, structured self-management can improve back-related disability and pain even when contact is remote, provided guidance is credible and actionable [56]. Our IPA adds that the form of delivery matters: viewers read pacing, camera framing, and explicit error-avoidance cues as signals of "safety to try," which appears to be a prerequisite for reframing AO from threat to opportunity. Clinically, briefing scripts that normalise initial caution, plus curated video libraries that begin with low-threat demonstrations and progressions, may help patients move from protective avoidance toward exploratory practice. Consistent with this interpretation, randomized evaluations of digitally delivered back-pain programs have shown that, when guidance is structured and credible, outcomes can be comparable to usual in-person physiotherapy, supporting the clinical utility of remote AO-supported self-management [57].

## Instructional delivery as the gateway to understanding

Across accounts, participants distinguished between understanding the movement and the quality of instruction. When instructions were concrete (goal clarity), richly cued (error-avoidance, breathing, posture), and paced for learning, action comprehension improved; when absent or rushed, comprehension faltered and avoidance scripts re-emerged. This dovetails with social–cognitive models of observational learning—attention, retention, and reproduction are instruction-sensitive processes [58]—and with neurocognitive evidence that AO engages distributed networks whose responses are tuned by task relevance and familiarity [20]. The practical takeaway is not that any AO will do, but that instructional craft (delivery and representation) is part of the clinical dose. Beyond clinical settings, platform dynamics also shape instructional uptake. Exposure to fitness content through social media can shift intentions and health behaviors via perceived credibility and modeling quality [59].

Our interpretative lens extends these results by showing that viewers map delivery features onto safety and competence signals. Where instructions were dense or rushed, CLBP participants reported breakdowns in "action

comprehension," prompting avoidance; where options and cues were explicit, they described an immediate uptick in "I can do this." That mechanism plausibly mediates adherence in digital programs.

### Contextual moderators: Prior pain, self-efficacy, and digital literacy

Participants' meaning-making was shaped by their pain history, perceived competence, and comfort navigating online media. Supportive evidence from recent digital back pain programmes shows that guidance intensity and credibility signals matter for engagement and outcomes [56]. In social platforms, perceived creator credibility and parasocial connection can amplify or dampen intentions to act on health content [37]. Our accounts suggest that CLBP participants scrutinise source credibility and adaptiveness (e.g., presence of modifications) more than asymptomatic viewers. Clinically, this argues for (i) clinicians providing short "why this is safe" primers before video use; (ii) level-tagging (beginner/intermediate/advanced) to match perceived competence; and (iii) supporting digital literacy (how to use playlists, captions, playback speed, and save functions).

### Cultural transferability and equity considerations

All interviews were conducted in Spain, and participants engaged primarily with Spanish-language or culturally familiar content. Video tone, body ideals, and instructional style vary by culture and may shift safety/credibility appraisals. Digital access and platform literacy also differ across age and socioeconomic groups, potentially widening gaps in who benefits from AO-based resources. We encourage cross-cultural adaptation (language, examples, models) and minimal-literacy design (onscreen captions, iconography, short segments), followed by local user-testing before broader rollout.

From a service-delivery perspective, health-system reviews of digital MSK services identify recurring implementation determinants—governance/guidelines, integration into clinical routines and information workflows, stakeholder acceptance/usability, and financing/incentives—that are directly pertinent to deploying AO videos in routine practice [60].

### Practice guidance derived from the IPA

Translating these qualitative insights, we propose three implementable tactics for AO-based exercise content in CLBP rehabilitation:

1. Safety-first scaffolding: Open videos with a one-sentence safety rationale and show an immediately "doable" version before harder variants; keep early exposures short to promote successful first attempts. This matches patient preferences here and complements supported self-management evidence.

2. Instructional minimalism with explicit goals: State the functional aim ("stand up with less bracing"), break the task into two or three observable steps, and add one error cue to avoid. This directly targets the action-comprehension gap described by participants and is consistent with current learning-from-others frameworks.

3. Contextual tailoring and credibility: Where possible, host or curate content under the clinic's channel/profile, add brief clinician voice-over or captions, and use platform tools (chapters, speed control) to reduce cognitive load. Evidence from digital health literacy and parasocial dynamics suggests these signals can increase intention to follow through. Moreover, parasocial bonds with creators can increase perceived self-efficacy and intention to enact exercise guidance, a mechanism especially relevant when clinicians curate or co-brand content [61]

### Strengths and limitations

This study's core strength is its IPA design, which privileges idiographic depth and participants' sense-making. By attending to how people with and without CLBP interpret AO exercise videos, the analysis adds explanatory granularity to "what works"

signals from intervention studies—clarifying why videos feel usable, for whom, and under what instructional conditions. Methodological rigor was supported through independent dual coding with negotiated consensus, maintenance of an audit trail, and reflexive practices (journaling, memoing, peer debriefing). Sampling targeted maximum variation (sex, age strata, CLBP duration/severity, digital literacy), and sample adequacy was justified via information power within an idiographic IPA frame.

Several limitations should be acknowledged. First, interview data capture subjective experience and intention; willingness to attempt exercises may not translate into behavior or functional change. Second, we did not directly observe motor performance alongside interviews, so interpretative shifts (e.g., reduced threat appraisal, better action understanding) cannot be causally linked to execution quality or outcomes. Third, the single-country sample and specific recruitment settings may limit cultural transferability and generalizability. Fourth, technology familiarity likely shaped participants' experiences with core video features (e.g., pausing, replay, playback speed), echoing implementation challenges reported for digital back-pain programs and apps [62]. Finally, as is typical in chronic pain research, current emotional state and prior pain history may have influenced recall and interpretation, adding heterogeneity to accounts; our reflexive procedures mitigate but do not eliminate this risk.

## Future directions

Design, evaluation, and implementation work should proceed in parallel. Design: co-design AO video libraries with patients and clinicians to operationalize a "safety-first, graded-options" template—clear goals, explicit safety messaging, beginner variants, pacing aligned with learning, and transparent credibility signals. Measurement: develop and validate a concise instrument that indexes the key experiential dimensions identified here (threat appraisal, perceived competence, instructional clarity, familiarity) to inform clinical tailoring and monitor change. Evaluation: test therapist-briefed, curated AO playlists against generic AO content in pragmatic hybrid trials; use mixed-methods to connect interpretative shifts (e.g., threat→opportunity reappraisal; comprehension gains) with downstream adherence, function, and pain-related disability, extending signals from supported self-management and digital implementations. Implementation: evaluate culturally localized curation strategies and clinician-supported onboarding to address digital-literacy variation and trust; examine implementation outcomes (acceptability, feasibility, fidelity, cost) and equity impacts across socioeconomic groups.

Longitudinal studies that embed AO resources into routine care—with periodic professional feedback, progression support, and data-governance safeguards—can test durability of engagement and clinical benefit. Where feasible, future qualitative work should be paired with direct observation of task performance or brief behavioral probes, allowing triangulation between sense-making and execution. Finally, given long-standing evidence that perspective and instructional delivery modulate observational learning, targeted experiments can isolate which delivery elements (e.g., safety cues, graded variants, pacing) most effectively translate interpretative clarity into confident enactment.. Furthermore, it would be relevant to investigate the impact of various types of videos and observational perspectives, given that the literature suggests that egocentric and allocentric perspectives can activate different neural networks and affect the efficacy of motor learning [63].

## Conclusion

This IPA indicates that adults with and without CLBP appraise video-based action observation through a layered process rather than as neutral stimuli. Participants first read delivery and representational features—goal clarity, explicit safety cues, pacing/tempo, viewpoint (egocentric/allocentric), and modeling fidelity/"like-me" exemplars—which shape action comprehension and perceived task demand. These cognitions, together with self-appraisal of capacity (competence and anticipated workload) and an observed shift in self-efficacy, inform a threat-versus-opportunity appraisal (fear/avoidance rules vs. motivation to try). Uptake occurs when videos feel understandable, safe, and adaptable (graded options/stepwise structure), and follow-through across days/weeks is enabled or hindered by symptoms, time, boredom, reminders, and available progressions; several participants reported functional gains even without pain change. Contextual

moderators—prior pain experiences, baseline self-efficacy, digital/platform literacy, and source credibility—consistently calibrated these pathways.

Practice implications are concrete: prioritise safety-first scaffolding and graded variants; make goals and one "error-to-avoid" cue explicit; choose viewpoint/pacing that reduce perceived demand; represent mixed-ability models to support "like-me" identification; and provide playlists with progressions, captions, and speed control to aid adherence. These findings specify design and implementation levers for AO resources in CLBP rehabilitation and provide targets for future mixed-methods and pragmatic evaluations.

## Supporting information

**S1 File. Semi-Structured Interview Guide.** The final version of the guide is presented, including the complete information on all items and probing questions.
(DOCX)

**S2 File. Audit Trail Summary.** This audit trail summarizes analytic decisions, consensus procedures, and reflexive memos across the seven-phase IPA workflow, aligned with COREQ/SRQR. It complements the Methods and provides a transparent account of how codes evolved into sub-themes and themes.
(DOCX)

**S1 Table. Codebook and Thematic Summary.** This table summarizes themes, sub-themes, analytic definitions, exemplar quotations, indicative counts by group, and the explicit linkage to delivery/representation features identified in the manuscript ("Instructional delivery as the gateway to understanding").
(DOCX)

**S1 Fig. Thematic pathway from video delivery/representation to uptake and adherence in action-observation exercise for CLBP: an IPA-derived map.** Proposed pathway linking **Delivery & Representation** (goal clarity, explicit safety cues, pacing/tempo, viewpoint, modeling fidelity/"like-me") to **Cognitive Processes** (attention, action comprehension, error-avoidance parsing) and **Self-Appraisal of Capacity** (perceived competence, anticipated workload). A **Self-efficacy shift** then tilts the **Appraisal** toward **Threat (fear, avoidance rules)** or **Opportunity (motivation to try)**, which in turn relates to **Intended Uptake & Adherence** (attempt, short-term use, ongoing adherence). **Contextual Moderators** (prior pain experiences, baseline self-efficacy, digital/platform literacy, source credibility, familiarity with action) act across stages. **Follow-through factors** (time, symptoms, reminders, boredom, progressions; functional changes even without pain change) influence maintenance. Arrows indicate directional influence within participants' sense-making rather than causal effects.
(TIF)

## Author contributions

**Conceptualization:** Roy La Touche, Alba Paris-Alemany, Joaquín Pardo-Montero.

**Data curation:** Roy La Touche, Prado Silván-Ferrero, Encarnación Nouvilas-Pallejá.

**Formal analysis:** Roy La Touche, Prado Silván-Ferrero, Encarnación Nouvilas-Pallejá, Alba Paris-Alemany, Miguel Ángel Sorrel, Joaquín Pardo-Montero.

**Funding acquisition:** Roy La Touche, Alba Paris-Alemany, Joaquín Pardo-Montero.

**Investigation:** Roy La Touche, José Vicente León-Hernández, Prado Silván-Ferrero, Encarnación Nouvilas-Pallejá, Alba Paris-Alemany, Miguel Ángel Sorrel, Joaquín Pardo-Montero.

**Methodology:** Roy La Touche, Alba Paris-Alemany, Miguel Ángel Sorrel, Joaquín Pardo-Montero.

**Project administration:** Roy La Touche, Alba Paris-Alemany.

**Resources:** Roy La Touche, José Vicente León-Hernández, Alba Paris-Alemany.

**Software:** Roy La Touche, Prado Silván-Ferrero, Encarnación Nouvilas-Pallejá, Miguel Ángel Sorrel, Joaquín Pardo-Montero.

**Supervision:** Roy La Touche, José Vicente León-Hernández, Alba Paris-Alemany, Miguel Ángel Sorrel, Joaquín Pardo-Montero.

**Validation:** Roy La Touche, José Vicente León-Hernández, Alba Paris-Alemany, Miguel Ángel Sorrel, Joaquín Pardo-Montero.

**Visualization:** Roy La Touche, Encarnación Nouvilas-Pallejá, Joaquín Pardo-Montero.

**Writing – original draft:** Roy La Touche, José Vicente León-Hernández, Prado Silván-Ferrero, Encarnación Nouvilas-Pallejá, Alba Paris-Alemany, Miguel Ángel Sorrel, Joaquín Pardo-Montero.

**Writing – review & editing:** Roy La Touche, José Vicente León-Hernández, Prado Silván-Ferrero, Encarnación Nouvilas-Pallejá, Alba Paris-Alemany, Miguel Ángel Sorrel, Joaquín Pardo-Montero.

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
