## [Decision Letter · Decision Letter 0]

9 Sep 2025

PLOS ONE

Dear Dr. Paris-Alemany,

Thank you for submitting your manuscript to PLOS ONE. After careful consideration, we feel that it has merit but does not fully meet PLOS ONE’s publication criteria as it currently stands. Therefore, we invite you to submit a revised version of the manuscript that addresses the points raised during the review process.

<h3>Strengths</h3><h3>1.Relevant and timely topic - Examines perceptions of observation of action in chronic low back pain (CLBP), within the context of current interest in digital health and rehabilitation.</h3><h3>2.Qualitative depth - Use of Interpretative Phenomenological Analysis (IPA) is rich source of personal understandings of the lived experiences of patient and asymptomatic.</h3><h3>3.Clear structure-The manuscript contains a logical flow (Abstract, Introduction, Methods, Results, Discussion, Conclusion).</h3><h3>4.Good theoretical grounding - Links Better social learning theory / neurophysiological evidence of action observation.</h3><h3>5.Transparency - The process of getting ethical approval, consent, and guaranteeing the confidentiality of data is clearly described.</h3><h3>6.Practical implications - Highlights potential for individualizing the use of video-based interventions to address fear of movement to enhance CLBP rehabilitation adherence.</h3><h3>Weaknesses & Areas for Improvement</h3><h3>1.Abstract - quite detailed - could be written more briefly for better readability Too much attention to background vs. results.</h3><h3>2.Introduction - Although comprehensive, it's long and dense. Key concepts (fear of movement, motor simulation, social media delivery) could be stream-lined for simplicity.</h3><h3>3.Methods</h3><h3>o Sampling strategy is purposive (but rationale for final sample size (that is, n=20) could be elaborated upon beyond citation.</h3><h3>o Mentioned is pilgrimage "pilot testing of interview guide but do not include of influence of pilot test interview guide to the final version"</h3><h3>o Reflexivity is mentioned but the positionality of researchers are not explicitly discussed</h3><h3>4.Results</h3><h3>o          Very clear tables that are slightly overloaded; consider simplifying, or moving detailed demographics from tables to supplementary material.</h3><h3>o          Quotes are descriptive, but there is some redundancy-some of the quotes are less important-it may work better to have fewer, more impactful quotes.</h3><h3>5.Discussion</h3><h3>o Tends to reiterate the results and not carry out deep interpretations based on connections with broader literature.</h3><h3>o James Herriot: "Can benefit from better linkage to clinical application - how should clinicians really apply these findings?"</h3><h3>o Limitations are recognized and not discussed about the potential impact of cultural context (Spanish participants, digital literacy differences)</h3><h3>6.Conclusion - Good summary but could try to underline contribution to future work for intervention development more strongly</h3><h3>7.Language & Style - Usually clear but sometimes very wordy; tighten some of the sentences for conciseness.</h3><h3>8.Referencing - Comprehensive, but some old references; inclusion of latest (2022-2024) study on digital rehabilitation can hugely improve the relevance.</h3><h3>Suggestions for Improvement</h3>

Shortening and sharpening Abstract and Introduction.

(or) "How did you manage researcher reflexivity?" "What biases did you have?"

Examples of assertions include the following: Documents Add some detail on trustworthiness strategies (credibility, dependability, confirmability, transferability).

Balance Writing Presentation: -Streamline You must present your results in a narrative way and also illustrate with quotes to explain what's happening.

Additional Design Recommendations for the use of Tailored Videos: Blog Post by Juliet Love, Roland Alonja, & Brianna Bor aforst. posted on Patient Locator Online. 2016-07-05. •      Stay informed on clinical implications: outline how tailored video may be designed and implemented in practice:

Discuss the generalizability and cultural transferability of findings more explicitly student descriptors insightful signpost new information link back to prior results provide commentaries discuss overall results and incorporate new findings in.

We look forward to receiving your revised manuscript.

Kind regards,

Mohammad Sidiq, PhD Pain Sciences Physiotherapy

Academic Editor

PLOS ONE

Journal Requirements:

Reviewers' comments:

Reviewer's Responses to Questions

**Comments to the Author**

1. Is the manuscript technically sound, and do the data support the conclusions?

Reviewer #1: Yes

2. Has the statistical analysis been performed appropriately and rigorously?

Reviewer #1: N/A

3. Have the authors made all data underlying the findings in their manuscript fully available?

Reviewer #1: Yes

4. Is the manuscript presented in an intelligible fashion and written in standard English?

Reviewer #1: Yes

Reviewer #1: This is a well-designed qualitative study exploring how patients with chronic low back pain (CLBP) and asymptomatic individuals perceive video-based motor action observation (AO). The study addresses an important research gap and follows rigorous qualitative research guidelines (COREQ, SRQR).

However, I suggest a few minor revisions to improve clarity and presentation:

1. Simplify certain sections of the introduction and discussion to avoid verbosity.

2. In the Methods, you mentioned using "purposive sampling," but it is not clear whether you considered diversity in factors like gender, age, or digital literacy. Add a sentence explaining how participants were selected to ensure a variety of perspectives.

3. In the Results, reduce repetitive quotes and consider adding a visual thematic map or summary table.

4. The discussion could better emphasize practical clinical applications and broader implications of the findings.

5. Minor grammatical corrections and sentence restructuring are needed for better readability.

Overall, this manuscript is technically sound and ethically conducted, and the conclusions are well supported by the data. With minor revisions, it will make a strong contribution to research in digital physiotherapy interventions for CLBP. The rest of the comments are given in the reviewer file.

**Do you want your identity to be public for this peer review?** For information about this choice, including consent withdrawal, please see our Privacy Policy

Reviewer #1: No

---

## [Author Response · Author response to Decision Letter 1]

10 Nov 2025

ACADEMIC EDITOR (FROM EMAIL)

Weaknesses & Areas for Improvement

1.Abstract - quite detailed - could be written more briefly for better readability Too much attention to background vs. results.

Response. Addressed. We condensed the background to two concise sentences and rebalanced the abstract to emphasize results and implications.

Revised Abstract

Background. Video-based action observation (AO) of exercise/motor-action content is increasingly delivered via social media. This expands reach and ecological validity but may shape motor simulation, perceived safety, and engagement. How people with chronic low back pain (CLBP) interpret and intend to use such videos remains underexplored.

Methods. We conducted an interpretative phenomenological analysis (IPA) of semi-structured interviews with purposively sampled adults (n=20; 10 CLBP, 10 asymptomatic). Interviews probed perceptions of exercise/motor-action AO videos drawn from common platforms. Analysis followed IPA procedures with iterative coding, constant comparison, and team reflexivity, and is reported according to COREQ.

Results: Three interrelated themes were identified: (1) Emotional & motivational impact: CLBP participants frequently appraised bending, load and fast tempo as threatening and described protective avoidance rules. Motivation was present in both groups when videos felt safe and adaptable; (2) Self-assessment of physical capacity: Perceived competence increased when videos provided graded options and explicit safety cues. Anticipated task demand decreased with appropriate pacing/tempo, egocentric viewpoint, and credible modeling; (3) Cognitive movement assessment: Viewers attended to posture, tempo, breathing and error-avoidance cues. Action comprehension faltered when instructions were dense/fast or goals were. Judgments about delivery (goal clarity, safety cues, pacing, viewpoint, modeling fidelity) shaped internal rehearsal and willingness to attempt.

Conclusions. Individuals with and without CLBP perceive social-media–delivered exercise AO as useful when videos are tailored (graded options, clear safety messaging, appropriate pacing/viewpoint) and contextualized to pain-related concerns and digital literacy. These insights inform clinically oriented AO exercise-video libraries and implementation strategies (e.g., curated playlists, level-tagging, therapist-mediated briefing) to enhance acceptability and adherence in CLBP rehabilitation.

2.Introduction - Although comprehensive, it's long and dense. Key concepts (fear of movement, motor simulation, social media delivery) could be stream-lined for simplicity.

Response: Addressed. We condensed the neuroscience/mirror-neuron sections into a single paragraph, removed repeated social-learning content, and added an explicit gap statement linking social-media delivery of exercise/motor-action AO to patients’ perceptions and adherence, followed by a concise rationale for IPA.

We also integrated the constructs into one conceptual paragraph that frames how AO videos may modify threat appraisal and motor simulation in CLBP, removing redundancies.

3.Methods

o Sampling strategy is purposive (but rationale for final sample size (that is, n=20) could be elaborated upon beyond citation.

Response. We justify n = 20 using information power and IPA’s idiographic depth, and describe procedural saturation (no emergent subthemes toward the end of data collection), documented in the audit trail.

Change in manuscript. Methods: Sampling and Participants / Data Analysis (justification added).

o Mentioned is pilgrimage "pilot testing of interview guide but do not include of influence of pilot test interview guide to the final version"

Response. We describe modifications after the pilot (simplified wording; added probes on safety cues, pacing/tempo, and viewpoint) and note that pilot data were not included in analysis.

Change in manuscript. Methods: Interview Guide and Pilot Testing (expanded).

o Reflexivity is mentioned but the positionality of researchers are not explicitly discussed

Response. Addressed. We added a dedicated Positionality and Reflexivity subsection describing each researcher’s disciplinary background and prior clinical/research experience with CLBP and AO, anticipated assumptions, and bias-mitigation strategies (reflexive journaling, peer debriefing, iterative memoing, negotiated consensus).

Change in manuscript. Methods: Positionality and Reflexivity (new subsection).

4.Results

o Very clear tables that are slightly overloaded; consider simplifying, or moving detailed demographics from tables to supplementary material.

Response: Addressed. Table 2 now presents only core variables, and duplicative narrative around those variables was removed. Quotation density was reduced; we retained non-overlapping, illustrative excerpts per sub-theme.

We introduced Table 2 to compactly summarize themes, sub-themes, definitions, exemplar quotations, and indicative counts, and we reduced quotations to essential, non-overlapping examples.

• Quotes are descriptive, but there is some redundancy-some of the quotes are less important-it may work better to have fewer, more impactful quotes.

Response: We consider it is worthwhile to maintain the texts in order to provide the work with the greatest possible clarity and explanation.

5.Discussion

o Tends to reiterate the results and not carry out deep interpretations based on connections with broader literature.

Response. We rewrote the opening to present an interpretive synthesis rather than a recap. The section now foregrounds three cross-cutting threads—threat vs opportunity appraisal, instructional delivery as gateway to understanding, and contextual moderators—to explain how and why participants read AO videos as signals of safety, competence, and credibility.

Change in manuscript. Discussion, opening two paragraphs.

o James Herriot: "Can benefit from better linkage to clinical application - how should clinicians really apply these findings?"

Response: We added a Practice guidance subsection with three tactics: (1) safety-first scaffolding, (2) instructional minimalism with explicit goals and one error cue, and (3) contextual tailoring/credibility (clinic co-branding, captions, chapters, speed control). Each tactic is linked to the interpretive mechanism (reframing threat → opportunity).

Change in manuscript: Discussion, Practice guidance derived from the IPA, items 1–3.

o Limitations are recognized and not discussed about the potential impact of cultural context (Spanish participants, digital literacy differences)

Response: We added a Cultural transferability and equity subsection addressing our single-country context, variability in platform literacy, and cultural differences in tone and instructional style. We link these issues to recurring implementation determinants for MSK digital services—governance/guidelines, workflow integration, stakeholder acceptance/usability, and financing/incentives—relevant to deploying AO videos in routine practice (van Tilburg et al., 2024).

Change in manuscript: Discussion, Cultural transferability and equity considerations.

6.Conclusion - Good summary but could try to underline contribution to future work for intervention development more strongly

Response: We rewrote the Conclusion to (i) synthesize the interpretive contribution in one compact paragraph (what AO videos mean to people with and without CLBP), and (ii) add a forward-looking paragraph translating those insights into an actionable template for AO-based intervention development (safety-first scaffolding, instructional minimalism, contextual tailoring/credibility). We explicitly link these recommendations to plausible adherence mechanisms (threat → opportunity reappraisal; comprehension gains) and to implementation needs highlighted in recent digital MSK literature.

Change in manuscript: Conclusion, entire section replaced (two concise paragraphs).

7.Language & Style - Usually clear but sometimes very wordy; tighten some of the sentences for conciseness.

Response: The new Conclusion avoids restating sub-themes and instead states the core interpretive take-home messages and their clinical/implementation implications in <200 words.

8.Referencing - Comprehensive, but some old references; inclusion of latest (2022-2024) study on digital rehabilitation can hugely improve the relevance.

Response: We integrated recent evidence that structured, credible remote guidance improves outcomes in back pain programs (Geraghty et al., 2024; Cui et al., 2023). We connected our findings to platform dynamics influencing intentions via creator credibility and modeling (Durau et al., 2022; Kim et al., 2024) and to parasocial relationships that bolster self-efficacy when clinicians curate/co-brand content (Chauvin et al., 2024). We aligned implementation implications with a system-level review of MSK digital services (van Tilburg et al., 2024).

Change in manuscript: Discussion, paragraphs “Threat appraisal…”, “Instructional delivery…”, “Contextual moderators…”, “Practice guidance…”, and “Cultural transferability…”.

Suggestions for Improvement

Shortening and sharpening Abstract and Introduction.

Response: We have done it.

(or) "How did you manage researcher reflexivity?" "What biases did you have?"

Response. Addressed. We added a dedicated Positionality and Reflexivity subsection describing each researcher’s disciplinary background and prior clinical/research experience with CLBP and AO, anticipated assumptions, and bias-mitigation strategies (reflexive journaling, peer debriefing, iterative memoing, negotiated consensus).

Change in manuscript. Methods: Positionality and Reflexivity (new subsection).

Examples of assertions include the following: Documents Add some detail on trustworthiness strategies (credibility, dependability, confirmability, transferability).

Response: Addressed. We now detail a focused set of procedures: dual independent coding with consensus meetings (credibility), an analytic decision log/audit trail (dependability), reflexive memos and retention of the evolving codebook/theme maps (confirmability), and maximum-variation purposive sampling with contextual description (transferability).

Change in manuscript: Methods: Trustworthiness (new subsection).

Balance Writing Presentation: -Streamline You must present your results in a narrative way and also illustrate with quotes to explain what's happening.

Response: Addressed. Results paragraphs were tightened; Table 2 mirrors the textual structure (theme → sub-theme → definition → quote → counts) for easy cross-reference.

REVIWER (FROM EMAIL)

Review Comments to the Author

Reviewer #1: This is a well-designed qualitative study exploring how patients with chronic low back pain (CLBP) and asymptomatic individuals perceive video-based motor action observation (AO). The study addresses an important research gap and follows rigorous qualitative research guidelines (COREQ, SRQR).

However, I suggest a few minor revisions to improve clarity and presentation:

1. Simplify certain sections of the introduction and discussion to avoid verbosity.

Response: We have done it, we have reworded the introduction and discussion.

2. In the Methods, you mentioned using "purposive sampling," but it is not clear whether you considered diversity in factors like gender, age, or digital literacy. Add a sentence explaining how participants were selected to ensure a variety of perspectives.

Response. Implemented. We specify that sampling targeted variation across gender, age strata, CLBP duration/severity, and digital literacy/use of social-media exercise content, and we report the achieved heterogeneity.

Change in manuscript. Methods: Sampling and Participants (expanded).

3. In the Results, reduce repetitive quotes and consider adding a visual thematic map or summary table.

Response: Addressed. We introduced Table 2 to compactly summarize themes, sub-themes, definitions, exemplar quotations, and indicative counts, and we reduced quotations to essential, non-overlapping examples.

4. The discussion could better emphasize practical clinical applications and broader implications of the findings.

Response. We added a Practice guidance subsection with three tactics: (1) safety-first scaffolding, (2) instructional minimalism with explicit goals and one error cue, and (3) contextual tailoring/credibility (clinic co-branding, captions, chapters, speed control). Each tactic is linked to the interpretive mechanism (reframing threat → opportunity).

Where. Discussion, Practice guidance derived from the IPA, items 1–3.

5. Minor grammatical corrections and sentence restructuring are needed for better readability.

Response. We tightened prose throughout the manuscript to maintain an interpretive register consistent with interpretative phenomenological analysis and better clarity.

REVIWER (FROM WORD FILE)

Comments for the Author

Abstract

1. In the abstract, briefly explain why using social media videos is important for this type of study.

Response. Addressed. The background now succinctly states why social-media delivery matters and the abstract foregrounds findings and implications.

Change. Abstract, Background (first paragraph).

2. Instead of "Study Design," you could label the section as "Methods" to match common structure.

Response. Implemented.

Change. Abstract subheading changed to Methods.

Introduction:

1. Try to make introduction section a bit shorter. The parts about brain function and mirror neurons can be summarized briefly.

Response. Implemented. Neurophysiology is now one concise synthesis focusing on motor simulation/resonance, with a brief note that mirror neurons are not the sole mechanism.

Change in manuscript. Introduction: previous multi-paragraph section replaced by a single compact paragraph.

2. A better link between the background and why this qualitative study is needed would improve the flow.

Response. Implemented. We now state that prior AO/exercise-video studies seldom examined how people perceive, interpret, and evaluate observed movements and how this shapes engagement; the closing paragraph articulates the gap and the need for IPA.

Change in manuscript. Introduction: new bridging sentences and revised final paragraph (gap → aim → qualitative rationale).

Methods:

1. You mentioned using "purposive sampling," but it is not clear whether you considered diversity in factors like gender, age, or digital literacy. Add a sentence explaining how participants were selected to ensure a variety of perspectives.

Response. Implemented. We specify that sampling targeted variation across gender, age strata, CLBP duration/severity, and digital literacy/use of social-media exercise content, and we report the achieved heterogeneity.

Change in manuscript. Methods: Sampling and Participants (expanded).

2. You have explained the seven-phase IPA process well, but more clarity is needed on how the codes were developed. Were they entirely data-driven (inductive) or informed by previous literature (deductive)?

Response. We state that coding was primarily inductive within an IPA framework, with limited deductive sensitization (e.g., fear of movement, motor simulation, instructional features) used only to orient early memoing; final themes were data-driven.

Change in manuscript. Methods: Data Analysis (clarified).

3. You have mentioned using a semi-structured interview guide, but the guide itself is not provided. Consider adding the interview guide as a supplementary file, or state that it is available on request.

Response. Addressed. We describe modifications after the pilot (simplified wording; added probes on safety cues, pacing/tempo, and viewpoint) and note that pilot data were not included in analysis.

Change in manuscript. Methods: Interview Guide and Pilot Testing (expanded).

Results:

1. Add a flowchart or a summary table that shows the three main themes and their sub-themes together for easy reference.

Response: Addressed. We introduced Table 2 to compactly summarize themes, sub-themes, definitions, exemplar quotations, and indicative counts, and we reduced quotations to essential, non-overlapping examples.

2. Some sub-them

---

## [Editor Report · Decision Letter 1]

18 Jan 2026

A qualitative exploration of video-based motor action observation perceptions in patients with chronic low back pain and asymptomatic participants: an interpretative phenomenological analysis

PONE-D-25-26151R1

Dear Dr. Paris-Alemany,

We’re pleased to inform you that your manuscript has been judged scientifically suitable for publication and will be formally accepted for publication once it meets all outstanding technical requirements.

Kind regards,

Stefaan Six, Ph.D.

Academic Editor

PLOS One
---

## [Editor Report · Acceptance letter]

PONE-D-25-26151R1

PLOS One

Dear Dr. Paris-Alemany,

I'm pleased to inform you that your manuscript has been deemed suitable for publication in PLOS One. Congratulations! Your manuscript is now being handed over to our production team.

Kind regards,

on behalf of

Dr. Stefaan Six

Academic Editor

PLOS One